# What Are Differences in Perceptions about Climate Technologies between Experts and the Public?

**Jaeryoung Song**

Center for External Affairs and Policy Cooperation, National Institute of Green Technology, Jung-gu, Seoul 04554, Republic of Korea; makingbetterworld@nigt.re.kr

**Abstract:** The Korean government announced it was carbon-neutral in October 2021. In addition, as a follow-up, the government maintains related laws and systems and invests in facilitating research and technology development extensively. Government investment is ultimately the people's taxes, and for laws and systems, the people's support and interest are of importance. This study investigated carbon-neutral technologies of an expert group and the public, technology acceptance, and recommendation intention. The results of the study are as follows. First, the goodness-of-fit of the proposed model and its structure had a positive impact on technology acceptance (recommendation) intention. Second, the result of a multi-group comparative analysis of the impacts on the acceptance of carbon-neutral technologies by the public and experts was overall at an acceptable level. Third, as for perceptions about climate technologies among experts and the public, it was found that there were differences in effort expectancy, facilitating conditions, and network effects between the expert group and the public.

**Keywords:** climate technology; carbon-neutral; technology acceptance model

## 1. Background

Climate change is perceived as the most important issue facing mankind and affects mankind in various environments (energy, water, food, etc.), and the need for an approach from an integrated perspective to resolve this is emphasized [1]. Major developed countries are scrambling to make a transition to a green economy and create a new growth momentum [2]. In the dimension of resolving the climate change issue, climate technologies are gradually emphasized, and Korea makes efforts to resolve the social problems that have close influences on people's lives, utilizing science and technology. The government expanded the roles of science and technology in resolving social problems through the "policy plan for solving social problems based on science and technology (2014–2018)" in 2013 and "The 2nd comprehensive plan for solving social problems based on the S&T (2018–2022)" in 2018 and has promoted practical efforts.

Damages occur due to climate change, for example, extreme weather and disasters [3], and since the Paris Agreement, the area has expanded as the dimension of climate change response considering climate change adaptation problems, financial affairs, economic development of developing countries, and future generations beyond greenhouse gas mitigation [4,5]. In this regard, the roles of climate technologies in resolving the climate change issue and adapting to climate change are gradually emphasized. In the dimension of response to climate change, there is a discussion that the corresponding technologies are as important as adaptability to climate change, and the corresponding technologies of climate change are the key means of greenhouse gas mitigation, which receives attention as the solution to the new climate system and important technology of greenhouse gas mitigation and treatment in international society [6]. As mentioned above, Korea is devoted to resolving social problems through science and technology, and investments increase to resolve social problems such as the environment and energy also in the green climate

technology. However, people do not feel the outcome, and there are insufficient studies that discovered social problems or considered the level of public perceptions in this area [7]. Addressing the challenge of climate change requires international collaboration based on the full implementation of policy on the global agenda, openness, mutual solidarity, and cooperation. Therefore, it is necessary to redefine public relations and public diplomacy on technology and innovation in response to the climate crisis [8].

The public, as defined by Grunig, is distinct from the crowd and the masses [9–12]. Unlike the masses, the public refers to a group formed around a particular issue or situation, which functions as a single system because they acquire and process the same information and produce the same behaviors. Grunig argues that when organizations seek to expand public support for a particular issue [10], they should attempt to increase the activity of the potential important public, even if they are not directly related to the issue, by employing communication strategies that are positive in direction. In recent communication research on profit organizations such as businesses, many studies have observed factors and derivative effects related to their activities [13,14]. However, research on the perception gap between experts and the public using specialized knowledge and technology on specific issues (especially technology acceptance) has not received much attention from a strategic communication perspective. Considering that most businesses emphasize technology and innovation and recognize that customer participation is an essential element [15,16], the role of the public and their perception of these issues can serve as a barometer for national policymaking as their relationship with businesses evolves and influences the development of technology and industry.

Strategic communication, in other words, public relations, is an essential way of tackling climate change and managing the risks involved [17]. Studies of risk awareness for climate change itself have been conducted actively; however, in the responsive dimension, discussions of perceptions about science and technology have been extremely rare. It is time to move beyond the conventional passive approach of finding solutions within the existing technology and explore an active response with high public (ordinary people) engagement to find novel solutions that can create innovative climate technology [18]. Thus, this study will examine climate technologies, that is, carbon-neutral-related technologies, applying the technology acceptance model (TAM) and demonstrate the impact on the acceptance of the real technologies.

## 2. Conceptualization: Technology Acceptance Model and Differences in Perceptions

### 2.1. Literature Review on Global Climate Change and Climate Technology Research

Climate change is not a recent topic: international discussions on climate change have been taking place since the first World Climate Conference in 1979, followed by the UNFCCC (1992), the Kyoto Protocol (1997), and the 2015 Paris Agreement [19]. International cooperation to address global warming and climate change entered a full-fledged phase with the entry into force of the Kyoto Protocol in 2005. The principles of the Kyoto Protocol, which adopted "common but differentiated responsibilities" (CBDR) as a guiding principle and imposed mitigation obligations on developed countries, were confirmed in subsequent climate agreements [20]. In contrast, the Paris Agreement, adopted in 2015, requires mitigation efforts from all countries, not just developed countries. In addition to submitting nationally determined contributions (NDCs) and long-term low-emission sustainable development strategies (LEDS) under the Paris Agreement, countries have also made a series of carbon-neutral declarations.

Climate change is not only being addressed at the national and international political level. Unlike the UNFCCC Framework Convention, which focuses on state actors, both the Kyoto Protocol and the Paris Agreement Decision specify the role of ordinary citizens in addressing climate change. The Kyoto Protocol includes a commitment to raise public awareness of climate change at the national level and facilitate public access to relevant information [19]. The Paris Agreement also emphasizes cooperation at multiple levels

involving various actors, with particular emphasis on public awareness, participation, and access to information [21].

In their study of strategic communication on climate change, Song et al. argue that effective responses to climate change require public engagement and citizen action [17]. To engage the public in climate action, it is important to examine measures at the international and national levels as well. However, empirical research on technology acceptance among the public and the professional community that deals with climate technology policy or develops technologies proposed in this paper is expected to have a significant positive impact on the use and diffusion of technologies.

The trends observed in studies related to climate change and technology acceptance models in business development are as follows: Kyriakopoulos' research on low-carbon energy technologies in the context of sustainable energy systems highlights the importance of public acceptance of energy systems based on renewables [22]; Khoza et al. present a gender-specific analysis of climate-smart agriculture adoption [23]; Mohr and Kühl's study on the acceptance of artificial intelligence in German agriculture reviews the application of technology acceptance models and the theory of planned behavior [24]; Toft et al. have developed and researched technology acceptance models, such as the Norm Activation Model (NAM), the theory of reasoned action (TRA), and the theory of planned behavior (TPB), to investigate consumer acceptance of smart grid technology [25].

### 2.2. Technology Acceptance Model and Unified Theory of Acceptance of Technology

The technology acceptance model (TAM) proposed by Davis is evaluated as a model that can systematically materialize people's technology acceptance behavior and has a high explanation ability to explain users' information technology acceptance and use behavior [26]. This theory is the one that has a theoretical basis on the theory of reasoned action [27] and is based on the psychological theory that deals with beliefs, attitudes, behavioral intentions, and actions concerning how users accept new technologies [28–31]. In addition, the TAM has merits in that it is easy to transform and extend the model, and it is suitable for dealing with technology acceptance phenomena in various ways [32]. For this reason, in the dimension of the users' new technology acceptance and use, it is discussed and utilized in various ways at home and abroad.

It has mainly been utilized to investigate the introduction and use of computer technology and systems, etc., as factors at play, and beliefs of perceived usefulness (PU) and perceived ease of use (PEOU) are the core dimension of the TAM. Davis defined perceived usefulness as a subjective belief that work performance and effect would increase with the use of the technology and perceived ease of use as a subjective belief that no extraordinary physical and mental effort would be required in the use of the technology [26]. Perceived usefulness and perceived ease of use have important impacts on the assessment of acceptance of technology and affect attitude and behavioral intention in use, leading to the actual use. In various studies utilizing these two concepts, its excellence was recognized [28], and Venkatesh and Davis presented an extended technology acceptance model (eTAM) by adding external factors to this [33].

The eTAM is an alternative model that made up for the criticism that the early TAM overlooked the impact of social influence on new technology acceptance. In the eTAM, as antecedent factors of perceived usefulness, social characteristics (e.g., subjective norm, social image, spontaneity) and cognitive tool process (e.g., job relevance, result demonstrability) were proposed [34]. Of the factors of social characteristics, the subjective norm is used most frequently, and the subjective norm is an individual's perception of the opinions of others and groups in action [27]. The subjective norm has a direct correlation with the intention to use new technology and affects perceived usefulness as well, for the authority of a person or group with a social influence can increase the usefulness and can have a positive impact on the intention to use accordingly [33]. However, in that it is based on the theory of reasoned action and the expectation theory, not a single theory, it has also been pointed out as an overuse [28,35].

Yet, the TAM has been utilized consistently due to its merit of usefulness in describing the use of new technology, and a theory of acceptance of technology is proposed, overcoming the threshold through a combination of various theories and going through an elaboration process. Venkatesh and Davis proposed the unified theory of acceptance and use of technology (UTAUT) [36], which has been reestablished based on the TAM, cognitive behavioral therapy, theory of planned behavior, innovation diffusion theory, etc. This theory proposes four factors, including performance expectancy, effort expectancy, social influence, and facilitating conditions, as new integrated variables with multiple theory levels. This is evaluated to better explain the acceptance and use of technology than the TAM previously proposed.

In the acceptance of innovation or new technology, the TAM has been utilized in various aspects, its effectiveness has been proven, and its model has been elaborated. This study would borrow the four variables of the UTAUT based on the previous studies and examine its impact by adding network effects and innovativeness proposed in the existing studies of technology acceptance based on the argument of Venkatesh et al. that it would be necessary to set additional external variables according to each technology field and environment [37].

### 2.2.1. Performance Expectancy

Performance expectancy is a concept similar to perceived usefulness proposed in the TAM [38], which is a belief that the use or application of a new innovative technology or product would help duty fulfillment or job performance and means a personal belief in a better result through the technology [36]. There may be an expectation for a better result in realizing carbon-neutral by the use of carbon-neutral technologies, and this study judged that this would affect the acceptance of carbon-neutral technologies (recommendation) intention to set the following hypothesis.

**Hypothesis 1.** *Performance expectancy would have a positive (+) impact on technology acceptance (recommendation) intention.*

### 2.2.2. Effort Expectancy

Effort expectancy [38], a concept similar to perceived usefulness, means that it is easy and convenient to use technology without difficulty, and it is the degree of ease in the acquisition and use of the technology or a belief in ease [36]. This study set a hypothesis that the acceptance of carbon-neutral technologies (recommendation) intention would increase when one feels the ease of carbon-neutral technologies based on the result of the existing studies that effort expectancy affected the intention to use.

**Hypothesis 2.** *Effort expectancy would have a positive (+) impact on technology acceptance (recommendation) intention.*

### 2.2.3. Social Influence

Social influence is a variable inferred from those proposed in technology acceptance-related studies, such as subjective norms, social factors, and image, etc., which is the degree to which people around feel that it is necessary to use a new system or technology, looking at the user of it [36]. This study would apply social influence on the carbon-neutral technologies area for examination. In other words, this study set the following hypothesis, feeling the big influence of carbon-neutral technologies and predicting that people would accept (recommend) the carbon-neutral technologies when recommended by the people around them (society, other researchers, academia, etc.).

**Hypothesis 3.** *Social influence would have a positive (+) impact on technology acceptance (recommendation) intention.*

### 2.2.4. Facilitating Conditions

Facilitating conditions, concerned with whether there is technological and organizational infrastructure in the use of new technology, is the degree of a belief that the environment in which the technology would be used or utilized has been created [36,39]. facilitating conditions are also an important variable to consider in the dimension of the acceptance of carbon-neutral technologies as a factor affecting technology acceptance. In other words, it is expected that the higher the degree of a belief that the infrastructure has been created concerning the R&D of carbon-neutral technologies, the higher the intention to accept (recommend) the carbon-neutral technologies. Thus, this study would test this through the following hypothesis.

**Hypothesis 4.** *Facilitating conditions would have a positive (+) impact on technology acceptance (recommendation) intention.*

### 2.2.5. Network Effects

Network effects mean that the greater the range of use of a technology or product or the number of people who use it, the greater the benefit or efficacy through the technology and the phenomenon in which its relative value increases [40,41]. As it is expected that these network effects would also appear in the carbon-neutral area, this study proposes the following research hypothesis, expecting that the actual intention to accept (recommend) the technology would increase when one feels that R&D or investment increases concerning carbon-neutral technologies.

**Hypothesis 5.** *The network factor would have a positive (+) impact on technology acceptance (recommendation) intention.*

### 2.2.6. Innovativeness

Innovativeness means that one would accept and experience a new technology faster than others [42]. It has been reported that the intention to use information technology and system increases when one recognizes technological innovativeness [43]. To examine innovativeness in this study, the acceptance of carbon-neutral technologies (recommendation) intention will increase when an individual has innovativeness for carbon-neutral technologies. Thus, the following hypothesis was established.

**Hypothesis 6.** *Innovativeness would have a positive (+) impact on technology acceptance (recommendation) intention.*

### 2.3. Construal Level Theory, Differences in Perceptions between the Public and Experts

According to the construal level theory, people differentially interpret something according to psychological distance, a subjective experience of that, and there is a difference in the construal level of it [44]. This is divided into temporal, empathetic, social, and stochastic distances, and the difference according to the construal level has been discussed in climate change-related studies.

Leiserowitz, Maibach, Roser-Renouf, and Smith noted that one tended to evaluate the personal risk that was lower than the social risk of climate change when one felt climate change as a distant future event, and it occurred at a distance, that is when one recognized a greater psychological distance [45]. Furthermore, in opposition, it was proven that one would recognize a greater risk of climate change when one had a closer psychological distance [46].

Along with this discussion, there have been several studies showing that experts and the public recognize a crisis or risk differentially and that the level of their recognition of the risk differs. Experts generally tend to recognize a risk depending on a statistical estimate; however, the public tends to associate the risk in connection to a problem or associate something remote from reality [47,48]. In a study of the cause of climate change,

differentiated perceptions were found as follows. Experts judged that climate change was caused by humans, while the public thought of it to be a natural phenomenon but not caused by humans [49]. Additionally, the difference was found in studies that discussed the perception of climate change between experts and the public. For example, in a study that examined if there would be a difference in the degree of recognition of climate change-related concepts between the persons interested and the public, it was proven that the public recognized the concepts less than experts did [50] and that the experts evaluated the recognition, interest, feeling, and severity of climate change greater than the public did [51]. Han, Kim, and Kim examined Koreans' recognition of the risks of climate change by experts and the public and found the result that there were clear differences in recognition of the causes for and risks of climate change, personal reaction competence, and the understanding of the policy between the two groups [52].

Like this, it has been proven that the awareness of the risk differs depending on the psychological distance to a certain problem and that there are differences in recognition of the problem of climate change between experts and the public. However, it is necessary to have common interests and perspectives on social problems for a change in recognition in resolving the social problems [53]. In this sense, it is judged that the examination of the dimension of perceptions of experts and the public would be an important task. Thus, this study will examine the differences in perceptions between the public and experts in carbon-neutral technologies. The following research questions were set to examine whether there would be an average of the recognition of technology acceptance in the recognition related to technology acceptance presented above presented between experts and the public and whether the relationship between variables would differ.

**Research Question 1**. Are there differences in the average of the main variables (performance expectancy, effort expectancy, social influence, facilitating conditions, network effects, innovativeness, technology acceptance (recommendation) intention) between the public and experts?

**Research Question 2**. Are there differences in perceptions about technology for carbon-neutral acceptance between the public and the expert group?

As can be seen in Figure 1, in order to solve the aforementioned hypotheses and research questions, a research model was designed.

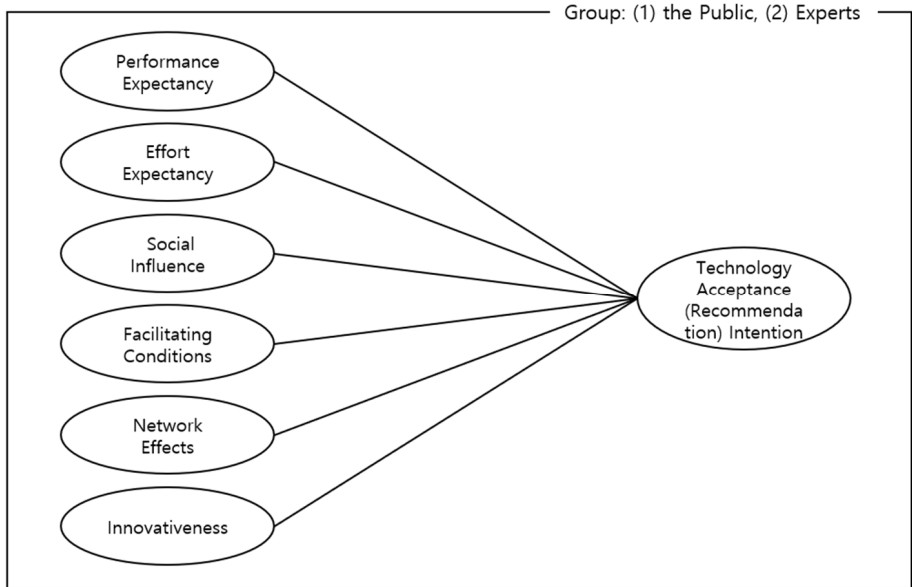

**Figure 1.** Research model.

## 3. Methodology

### 3.1. Operational Definition and Measurement of Variables

To borrow the related variables and examine their impacts on technology acceptance (recommendation) intention based on the existing studies of technology acceptance, the main variables were defined, and measurement items were established.

First, performance expectancy is the expectancy that utilizing new or innovative technology would help improve job or business performance, and this study examined that carbon-neutral technologies would contribute to carbon-neutral-related R&D, industry, and the market. Regarding the measurement items, six items, including "carbon-neutral technologies would overall contribute to carbon-neutral-related R&D achievements", "carbon-neutral technologies would overall increase productivity in related industries", and "carbon-neutral technologies would be useful in the low carbon, green industry", were measured on a 5-point Likert scale. Second, effort expectancy is the degree of ease of acquisition and use of technology or the belief in ease, and this study measured it, dividing it into the ease of industry application of carbon-neutral technologies, ease of commoditization, ease of management, and ease of convergence/integration grafting, etc. Concretely, there were five items, including "It would be easy to apply carbon-neutral technologies to the related industries", "It would be easy to manage carbon-neutral technologies in the related industries", and "carbon-neutral technologies would contribute to the fast realization of carbon-neutral." Next, social influence is users' subjective awareness of belief that a new technology/system should be used, and this study examined that through whether the carbon-neutral technologies were recommended by the surrounding people, industrial circles (company), and society and how big its influence was. Four items were measured, including "My co-workers would encourage the use of these technologies" and "Researchers of the convergence of these technologies would influence me." Fourth, facilitating conditions mean recognition of resources, infrastructure, and support for the use of technology, and this study measured the degree of participation in educational programs, the degree of shared vision, the degree of convergence, and comprehensive cooperation, concerning the R&D of carbon-neutral technologies, and set a total of five items by composing three items, including "Currently, industry-university-academy collaboration or convergence and complex cooperation between other fields for the R&D of carbon-neutral technologies have been revitalized", "I actively participate in capacity building or retraining programs for the R&D of carbon-neutral technologies." Fifth, network effects are concerned with whether the benefits and efficacy secured by using the relevant product are promoted as the number of users increases, which were measured with five items on the increase in joint research groups, the increase in R&D investments and projects, and the degree of diffusion into industrial circles. "Carbon-neutral technologies-related international joint research groups are increasing" and "carbon-neutral-related industry-university-academy collaboration R&D competition spreads" were included. Sixth, innovativeness was defined as the intention to openly accept a new technology or service, which consisted of six items on the innovativeness of carbon-neutral technologies, the innovativeness of industry–university–academy collaboration, and the recognition of innovativeness (e.g., "These technologies will break existing high-carbon industry-oriented practices" and "These technologies will promote the development of low-carbon new products and new services").

Lastly, "the intention to accept (recommend) carbon-neutral technologies" that comes under the intention to use is the intention to perform carbon-neutral technologies-related R&D or accept it, which consisted of six items, including "I will propose the active development of a promising technology to the national assembly, government, research society, and research foundation" and "I will actively recommend industrial circles the transfer and commercialization of related patents."

To the items difficult for the public to respond of the above ones, they were asked to respond, assuming that they were working in the carbon-neutral or environmental field.

### 3.2. Selection of Samples and Their Characteristics

For this study, the recognition of carbon-neutral technologies by experts and the public residing in Korea was surveyed respectively by commissioning a research company. The survey with experts was conducted with 238 persons from 19 October through 1 November 2021. Here, the experts employed in climate change-related technologies and policies were selected. The survey with the public used effective samples of 200 persons from 15 July through 19 July 2022, in collection and research. The demographic characteristics of the public and experts collected through the survey are presented in Tables 1 and 2.

**Table 1.** The public's demographic characteristics.

| | Classification | | Frequency (n) | Percent (%) |
|---|---|---|---|---|
| Sex | Male | | 95 | 47.5 |
| | Female | | 105 | 52.5 |
| Age | 20–29 | | 24 | 12.0 |
| | 30–39 | | 44 | 22.0 |
| | 40–49 | | 49 | 24.5 |
| | 50–59 | | 36 | 18.0 |
| | 60- | | 47 | 23.5 |
| Residence | Capital area and metropolitan cities | Seoul | 27 | 13.5 |
| | | Busan | 9 | 4.5 |
| | | Daegu | 7 | 3.5 |
| | | Incheon | 8 | 4.0 |
| | | Gwangju | 4 | 2.0 |
| | | Daejeon | 4 | 2.0 |
| | | Ulsan | 3 | 1.5 |
| | | Gyeonggi-do | 38 | 19.0 |
| | Other provincial areas | Gangwon-do | 10 | 5.0 |
| | | Chungbuk | 10 | 5.0 |
| | | Chungnam | 15 | 7.5 |
| | | Jeonbuk | 12 | 6.0 |
| | | Jeonnam | 11 | 5.5 |
| | | Kyeongbuk | 17 | 8.5 |
| | | Gyeongnam | 21 | 10.5 |
| | | Jeju-do | 4 | 2.0 |
| Educational background | High school graduation | | 45 | 22.5 |
| | University graduation | | 132 | 66.0 |
| | Master's degree | | 19 | 9.5 |
| | Doctoral degree | | 4 | 2.0 |
| | Total | | 200 | 100.0 |

To examine the respondents' characteristics, first, in the public, 95 (47.5%) were men, 105 (52.5%) were women, and 24.5% (n = 49) were in their 40s, followed by those in their 60s (n = 47, 23.5%); 30s (n = 44, 22.0%); 50s (n = 36, 18.0%); and 20s (n = 24, 12.0%). By residence, a survey was conducted by random allocation of residents in the capital area and metropolitan cities and those in other provincial areas at the ratio of 1:1, and the questionnaire was collected from 100 persons (50.0%) each. By the educational background of respondents in the public, 132 persons (66.0%) graduated from university, which took up more than half, followed by high school graduates, 22.5% (n = 45); master's degree holders and doctoral degree holders 19 (9.5%) and 4 (2.0%), respectively.

Next, in the expert group, there were more men than women, 198 men (79.4%) and 49 women (20.6%); most of them were in their 30s, 35.3% (n = 84), and the fewest were in their 20s, 3.4% (n = 8). By the place of work, 224 persons (94.1%) worked in Korea, and 14 persons (5.9%) were overseas workers. Of the experts, 109 persons (45.8%) worked at a government-funded research center, followed by a private enterprise, 21.0% (n = 50), a public institution, 18.5% (n = 44), and a university, 10.5% (n = 25). By duty, 83 persons

(34.9%) worked in R&D, followed by policy research, 77 (32.4%); business management, 41 (17.2%); and research administration, 19 (8.0%). R&D areas were various, with the energy transition sector, the most at 42.4% (n = 35), and the CCUS sector, the least at 4.8% (n = 4). The work experience in climate/environment/energy-related fields was evenly between less than 5 years and more than 12 years. A total of 78 persons (32.8%) worked for fewer than 5 years, 79 (33.2%) for fewer than 6 to 11 years, and 81 (34.0%) for more than 12 years. Lastly, 56.3% (n = 134) of the experts completed a doctoral course, which took up more than half.

**Table 2.** Experts' demographic characteristics.

| Classification | | Frequency (n) | Percent (%) |
|---|---|---|---|
| Sex | Male | 198 | 79.4 |
| | Female | 49 | 20.6 |
| Age | 20–29 | 8 | 3.4 |
| | 30–39 | 84 | 35.3 |
| | 40–49 | 74 | 31.1 |
| | 50–59 | 44 | 18.5 |
| | 60- | 28 | 11.7 |
| Place of Work | Domestic | Seoul | 72 | 30.3 |
| | | Busan | 1 | 0.4 |
| | | Daegu | 1 | 0.4 |
| | | Incheon | 4 | 1.7 |
| | | Gwangju | 3 | 1.3 |
| | | Daejeon | 53 | 22.3 |
| | | Ulsan | 7 | 2.9 |
| | | Gyeonggi-do | 30 | 12.6 |
| | | Gangwon-do | 2 | 0.8 |
| | | Chungbuk | 3 | 1.3 |
| | | Chungnam | 6 | 2.5 |
| | | Jeonbuk | 5 | 2.1 |
| | | Jeonnam | 2 | 0.8 |
| | | Kyeongbuk | 1 | 0.4 |
| | | Gyeongnam | 7 | 2.9 |
| | | Jeju-do | 5 | 2.1 |
| | | Sejong | 22 | 9.2 |
| | Overseas | Germany | 4 | 1.7 |
| | | U.K. | 4 | 1.7 |
| | | France | 4 | 1.7 |
| | | Other European Countries | 2 | 0.8 |
| Type of Institution | Government-funded research center | 109 | 45.8 |
| | Private enterprise | 50 | 21.0 |
| | Public institution | 44 | 18.5 |
| | University | 25 | 10.5 |
| | Other | 10 | 4.2 |
| Duty | R&D | 83 | 34.9 |
| | Policy research | 77 | 32.4 |
| | Business management | 41 | 17.2 |
| | Research administration | 19 | 8.0 |
| | Other | 18 | 7.6 |
| R&D areas | Energy transition sector | 35 | 42.2 |
| | Energy efficiency sector | 12 | 14.5 |
| | Digitization sector | 10 | 12.0 |
| | Low-carbon sector | 9 | 10.8 |
| | CCUS sector | 4 | 4.8 |
| | Other | 13 | 15.7 |
| Work experience in climate/environment/energy-related areas | Less than 5 years | 78 | 32.8 |
| | Less than 6 to 11 years | 79 | 33.2 |
| | More than 12 years | 81 | 34.0 |
| Educational background | University graduation | 31 | 13.0 |
| | Master's degree | 73 | 30.7 |
| | Doctoral degree | 134 | 56.3 |
| Total | | 238 | 100.0 |

*3.3. Analysis Method*

Basic analysis and average comparative analysis (*t*-test) were conducted using SPSS 26.0. In addition, to understand the structure of the technology acceptance model in the acceptance of carbon-neutral technologies, a confirmatory factor analysis and structural

equation modeling were performed, employing AMOS 23.0. In addition, a multi-group analysis was conducted to examine the differences between the experts and the public.

## 4. Results and Discussion

### 4.1. Mean, Standard Deviation, and Reliability of the Main Variables

The mean, standard deviation, and reliability of the main variables employed in this study are analyzed and presented in Table 3. To examine them overall, the reliability of the main variables is 0.70, which shows that the items are suitable for measuring each variable. The mean of the main variables is in the mid-threes. The mean of effort expectancy was the lowest at 3.51 (SD = 0.67) while that of recommendation intention was the highest at 3.94 (SD = 0.64).

**Table 3.** Mean, standard deviation, and reliability of the main variables.

| Variable | No. of Items | Mean | Standard Deviation | Reliability |
|---|---|---|---|---|
| Performance expectancy | 6 | 3.81 | 0.58 | 0.82 |
| Effort expectancy | 5 | 3.51 | 0.67 | 0.85 |
| Social influence | 4 | 3.73 | 0.63 | 0.76 |
| Facilitating conditions | 5 | 3.61 | 0.76 | 0.84 |
| Network effects | 5 | 3.77 | 0.59 | 0.79 |
| Innovativeness | 6 | 3.83 | 0.56 | 0.85 |
| Technology acceptance (recommendation) intention | 6 | 3.94 | 0.64 | 0.90 |

### 4.2. Test of the Goodness-of-Fit and Validity of the Measurement Tools

The results of the test of the goodness-of-fit and validity of the measurement tools are presented in Tables 4 and 5.

**Table 4.** Goodness-of-fit of the measurement model.

| Absolute Fit Index | | | | Comparative Fit Index | | |
|---|---|---|---|---|---|---|
| Chi-Square (df) | RMSEA | RMR | GFI | IFI | TLI | CFI |
| 1380.42 (608) *** | 0.05 | 0.04 | 0.84 | 0.90 | 0.89 | 0.90 |

*** $p < 0.001$.

First, the significance level of the chi-square value, the absolute fit index of goodness-of-fit was a little smaller than 0.001. RMESA and RMR were appropriate at 0.05 and 0.04, respectively. The GFI was somewhat lower than 0.90, the standard value; however, other indices were excellent. Thus, the overall absolute fit indices were at an acceptable level. The incremental fit indices, IFI, TLI, and CFI, were 0.90, 0.89, and 0.90, respectively. Thus, the incremental fit indices of the variables employed in this study are appropriate as they were higher than or approximate to the standard value, 0.90.

Next, in the result of the confirmatory factor analysis, the standardization factor load was greater than 0.60 in all variables, and the *p*-value of them was smaller than 0.001 in all variables. In addition, the average variance extracted (AVE) of all variables was greater than the standard value, 0.50, and the latent reliability was greater than 0.70 in all variables. Thus, all measurement variables (items) are statistically valid.

### 4.3. Correlations among the Main Variables

The correlations among the main variables employed in this study are identified and presented in Table 6. As a result of an analysis, the correlations among all variables were statistically significant based on $p < 0.001$. Effort expectancy had the highest correlation with performance expectancy (r = 0.73, $p < 0.001$), while it had the lowest correlation with facilitating conditions (r = 0.41, $p < 0.001$). Effort expectancy also had the lowest correlation

with facilitating conditions at r = 0.37 (*p* < 0.001). Social influence had the highest correlation with performance expectancy (r = 0.64, *p* < 0.001), while it had the lowest correlation with network effects (r = 0.47, *p* < 0.001). Facilitating conditions had high correlations with innovativeness and recommendation intention at r = 60 and 59 (*p* < 0.001), respectively. Network effects had a greater correlation with innovativeness compared to that with other variables (r = 0.65, *p* < 0.001), while it had the lowest correlation with facilitating conditions (r = 0.39, *p* < 0.001). Innovativeness also had a relatively lower correlation with facilitating conditions at r = 0.43 (*p* < 0.001), and technology acceptance (recommendation) intention had the highest correlation (r = 0.61, *p* < 0.001) with innovativeness and the lowest correlation with effort expectancy (r = 0.42, *p* < 0.001).

**Table 5.** Confirmatory factor analysis of the main variables.

| Latent Factor | Measurement Variable | B | β | S.E, | C.R (p) | AVE | Latent Reliability |
|---|---|---|---|---|---|---|---|
| Performance expectancy | Performance expectancy 1 | 1 | 0.69 | fix | | 0.55 | 0.88 |
| | Performance expectancy 2 | 0.96 | 0.64 | 0.08 | 12.37 *** | | |
| | Performance expectancy 3 | 1.08 | 0.69 | 0.08 | 13.28 *** | | |
| | Performance expectancy 4 | 1.17 | 0.67 | 0.09 | 12.97 *** | | |
| | Performance expectancy 5 | 1.04 | 0.60 | 0.09 | 11.55 *** | | |
| | Performance expectancy 6 | 0.98 | 0.67 | 0.08 | 12.87 *** | | |
| Effort expectancy | Effort expectancy 1 | 1 | 0.68 | fix | | 0.61 | 0.89 |
| | Effort expectancy 2 | 1.01 | 0.69 | 0.08 | 12.92 *** | | |
| | Effort expectancy 3 | 1.06 | 0.75 | 0.08 | 13.74 *** | | |
| | Effort expectancy 4 | 1.16 | 0.74 | 0.09 | 13.63 *** | | |
| | Effort expectancy 5 | 1.10 | 0.79 | 0.08 | 14.36 *** | | |
| Social influence | Social influence 1 | 1 | 0.64 | fix | | 0.54 | 0.83 |
| | Social influence 2 | 1.15 | 0.66 | 0.10 | 11.47 *** | | |
| | Social influence 3 | 1.08 | 0.63 | 0.10 | 10.99 *** | | |
| | Social influence 4 | 1.24 | 0.74 | 0.10 | 12.44 *** | | |
| Facilitating conditions | Facilitating conditions 1 | 1 | 0.81 | fix | | 0.54 | 0.85 |
| | Facilitating conditions 2 | 0.68 | 0.69 | 0.05 | 14.67 *** | | |
| | Facilitating conditions 3 | 0.77 | 0.69 | 0.05 | 14.66 *** | | |
| | Facilitating conditions 4 | 0.96 | 0.79 | 0.06 | 17.04 *** | | |
| | Facilitating conditions 5 | 0.68 | 0.60 | 0.06 | 12.51 *** | | |
| Network effects | Network effects 1 | 1 | 0.62 | fix | | 0.55 | 0.86 |
| | Network effects 2 | 0.93 | 0.58 | 0.09 | 9.88 *** | | |
| | Network effects 3 | 1.12 | 0.66 | 0.10 | 10.89 *** | | |
| | Network effects 4 | 1.23 | 0.75 | 0.10 | 11.95 *** | | |
| | Network effects 5 | 1.22 | 0.68 | 0.11 | 11.24 *** | | |
| Innovativeness | Innovativeness 1 | 1 | 0.71 | fix | | 0.63 | 0.91 |
| | Innovativeness 2 | 0.85 | 0.70 | 0.06 | 13.73 *** | | |
| | Innovativeness 3 | 0.91 | 0.74 | 0.06 | 14.34 *** | | |
| | Innovativeness 4 | 0.81 | 0.66 | 0.06 | 12.87 *** | | |
| | Innovativeness 5 | 0.93 | 0.73 | 0.07 | 14.21 *** | | |
| | Innovativeness 6 | 0.80 | 0.64 | 0.06 | 12.61 *** | | |
| Technology acceptance (recommendation) intention | Intention 1 | 1 | 0.76 | fix | | 0.71 | 0.94 |
| | Intention 2 | 1.03 | 0.79 | 0.06 | 16.83 *** | | |
| | Intention 3 | 1.05 | 0.79 | 0.06 | 17.01 *** | | |
| | Intention 4 | 1.02 | 0.77 | 0.06 | 16.57 *** | | |
| | Intention 5 | 1.06 | 0.79 | 0.06 | 16.84 *** | | |
| | Intention 6 | 0.98 | 0.74 | 0.06 | 15.78 *** | | |

*** *p* < 0.001.

**Table 6.** The correlation coefficients among the main variables.

| | 1 | 2 | 3 | 4 | 5 | 6 | 7 |
|---|---|---|---|---|---|---|---|
| 1. Performance expectancy | 1 | | | | | | |
| 2. Effort expectancy | 0.73 *** | 1 | | | | | |
| 3. Social influence | 0.64 *** | 0.60 *** | 1 | | | | |
| 4. Facilitating conditions | 0.41 *** | 0.37 *** | 0.52 *** | 1 | | | |
| 5. Network effects | 0.56 *** | 0.49 *** | 0.47 *** | 0.39 *** | 1 | | |
| 6. Innovativeness | 0.71 *** | 0.60 *** | 0.60 *** | 0.43 *** | 0.65 *** | 1 | |
| 7. Technology acceptance (recommendation) intention | 0.54 *** | 0.42 *** | 0.59 *** | 0.51 *** | 0.46 *** | 0.61 *** | 1 |

*** *p* < 0.001.

### 4.4. Test of Mean Difference in the Main Variables by Group (Experts, the Public)

To examine if there are differences in the mean of the key factors of the TAM, performance expectancy and effort expectancy, and their antecedent factors and recommendation

intention between the experts and the public, an independent-samples *t*-test was conducted, and the result is presented in Table 7. As a result of the analysis, there were differences in effort expectancy, facilitating conditions, and network effects between the expert group and the public.

**Table 7.** Test of mean difference in the main variables by group (*t*-test).

| Variable | Group | *N* | *M* | *SD* | *t* (*p*) |
|---|---|---|---|---|---|
| Performance expectancy | The public | 200 | 3.78 | 0.57 | −1.15 (0.251) |
| | Experts | 238 | 3.84 | 0.58 | |
| Effort expectancy | The public | 200 | 3.58 | 0.61 | 1.91 * |
| | Experts | 238 | 3.45 | 0.72 | |
| Social influence | The public | 200 | 3.77 | 0.62 | 1.26 (0.209) |
| | Experts | 238 | 3.70 | 0.64 | |
| Facilitating conditions | The public | 200 | 3.94 | 0.63 | 9.15 *** |
| | Experts | 238 | 3.33 | 0.75 | |
| Network effects | The public | 200 | 3.66 | 0.59 | −3.39 *** |
| | Experts | 238 | 3.85 | 0.58 | |
| Innovativeness | The public | 200 | 3.79 | 0.54 | −1.60 (0.111) |
| | Experts | 238 | 3.87 | 0.58 | |
| Technology acceptance (recommendation) intention | The public | 200 | 3.93 | 0.63 | −0.47 (0.641) |
| | Experts | 238 | 3.96 | 0.64 | |

* $p < 0.05$, *** $p < 0.001$.

First, for effort expectancy, the mean of the public was 3.58 (SD = 0.61), while that of experts was 3.45 (SD = 0.72), and t value was 1.906, and the significance level (p) was smaller than 0.05. Thus, statistically, the public evaluated the ease of carbon-neutral technologies higher than experts did. Second, also for facilitating conditions, the significance level of t value between the two groups was statistically significant (t = 9.15, $p < 0.001$), and the mean of the public was 3.94 (SD = 0.63), while that of experts was 3.33 (SD = 0.075). The public recognized facilitating conditions more for carbon-neutral technologies. Lastly, for network effects, at a statistically significant level, the mean of experts (M = 3.85, SD = 0.58) was greater than that of the public (M = 3.66, SD = 0.59) (t = −3.39, $p < 0.001$).

*4.5. Goodness-of-Fit of the Structural Model and Path Analysis*

The goodness-of-fit and structure of the model presented in this study are analyzed and presented in Tables 8 and 9. In the result, first, the goodness-of-fit of the structural model was overall excellent. The chi-squared value, the absolute fit index, was 1417.20 ($p < 0.001$), which was statistically significant; RMESA, 0.06; RMR, 0.04; CFI, the representative incremental fit index, 0.90; and TLI, 0.89.

**Table 8.** Goodness-of-fit of the structure model.

| Absolute Fit Index | | | | Comparative Fit Index | | |
|---|---|---|---|---|---|---|
| Chi-Square (df) | RMSEA | RMR | GFI | IFI | TLI | CFI |
| 1417.20 (580) *** | 0.06 | 0.04 | 0.85 | 0.91 | 0.89 | 0.90 |

*** $p < 0.001$.

Next, the result of the path analysis, the causal relations of variables are as follows. All variables other than network effects β = 0.03 ($p > 0.05$) had positive impacts on technology acceptance (recommendation) intention. Of them, the impact of performance expectancy on technology acceptance (recommendation) intention was the greatest β = 0.54 ($p < 0.001$), followed by the effect of effort expectancy, β = 0.40 ($p < 0.001$). The impact of social influence

was $\beta = 0.27$ ($p < 0.001$); that of facilitating conditions, $\beta = 0.18$ ($p < 0.001$); and that of innovativeness, $\beta = 0.38$ ($p < 0.001$) on technology acceptance (recommendation) intention.

**Table 9.** Result of path analysis.

| Path | | B | S.E. | $\beta$ | C.R. | $p$ |
|---|---|---|---|---|---|---|
| Performance expectancy | | 0.60 | 0.04 | 0.54 | 13.43 | *** |
| Effort expectancy | | 0.40 | 0.04 | 0.42 | 9.77 | *** |
| Social influence | Technology acceptance | 0.27 | 0.05 | 0.27 | 5.71 | *** |
| Facilitating conditions | (recommendation) intention | 0.18 | 0.04 | 0.22 | 5.27 | *** |
| Network effects | | 0.04 | 0.05 | 0.03 | 0.72 | 0.471 |
| Innovativeness | | 0.38 | 0.06 | 0.33 | 6.58 | *** |

*** $p < 0.001$.

*4.6. Comparative Analysis between Groups*

To compare the difference in the impact on the acceptance of carbon-neutral technologies between the public and experts, the results of multiple group comparative analysis are presented in Tables 10 and 11. First, to examine the goodness-of-fit of the multi-group comparison model, it was overall at an acceptable level (See Table 10). In the result of the test of the difference between the two groups, the difference in the chi-square was 89.36 ($\Delta$df = 37), and the significance level was 0.000. Thus, the difference was statistically significant based on $p < 0.001$.

**Table 10.** Multi-group comparison model goodness-of-fit.

| | Absolute Fit Index | | | | Comparative Fit Index | | |
|---|---|---|---|---|---|---|---|
| | Chi-Square (df) | RMSEA | RMR | GFI | IFI | TLI | CFI |
| Unconstrained model | 2232.03 (1224) *** | 0.04 | 0.05 | 0.78 | 0.88 | 0.87 | 0.88 |
| Constrained model | 2321.39 (1261) *** | 0.04 | 0.05 | 0.77 | 0.88 | 0.87 | 0.88 |

*** $p < 0.001$.

**Table 11.** Comparison of the public model and the expert model.

| Classification | $\Delta$chi-Square | $\Delta df$ | $p$ |
|---|---|---|---|
| Unconstrained model—Constrained model | 89.36 | 37 | 0.000 |

The result of path analysis is like Table 12, and the major difference is as follows. First, the impact of performance expectancy on technology acceptance (recommendation) intention was greater in the public than among experts. Specifically, the impact was $\beta = 0.69$ ($p < 0.001$) in the public and $\beta = 0.42$ ($p < 0.001$) in the expert group. Second, for effort expectancy, the impact on technology acceptance (recommendation) intention was greater in the public, $\beta = 0.49$, ($p < 0.001$) than in the expert group ($\beta = 0.39$, $p < 0.001$). The difference in facilitating conditions between the two groups was the greatest, and the impact of facilitating conditions on technology acceptance (recommendation) intention was $\beta = 0.55$ ($p < 0.001$) in the public and $\beta = 0.16$ ($p < 0.001$) in the expert group. Meanwhile, the impacts of social influence and innovativeness on technology acceptance (recommendation) intention were greater in the expert group. In the public, social influence affected technology acceptance (recommendation) intention at $\beta = 0.17$ ($p < 0.01$); however, the impact was $\beta = 0.26$ ($p < 0.001$) in the expert group. The impact of innovativeness on technology acceptance (recommendation) intention was $\beta = 0.25$ ($p < 0.001$) in the public and $\beta = 0.28$ ($p < 0.001$) in the expert group, somewhat greater in the expert group though not a big difference.

**Table 12.** Comparison of the path between the public and experts.

| Path | | The Public | | Experts | |
|---|---|---|---|---|---|
| | | $\beta$ | $p$ | $\beta$ | $p$ |
| Performance expectancy | | 0.69 | *** | 0.42 | *** |
| Effort expectancy | | 0.49 | *** | 0.39 | *** |
| Social influence | Technology acceptance | 0.17 | 0.002 | 0.26 | *** |
| Facilitating conditions | (recommendation) intention | 0.55 | *** | 0.16 | 0.015 |
| Network effects | | −0.01 | 0.881 | 0.03 | 0.713 |
| Innovativeness | | 0.25 | *** | 0.28 | *** |

*** $p < 0.001$.

## 5. Conclusions and Future Direction

In the results of an analysis of the goodness-of-fit and structure of the model presented in this study, all variables except for network effects had positive impacts on technology acceptance (recommendation) intention at a statistically significant level. Of them, performance expectancy and effort expectancy were the greatest, and social influence, facilitating conditions, and innovativeness affected technology acceptance (recommendation) intention at a certain level. It is expected that all respondents think that carbon-neutral technologies help promote duty fulfillment and job performance and recognize the convenience and ease of the technologies at a high level. In addition, to compare the difference in the impact on the acceptance of carbon-neutral technologies between the public and experts, it was overall at an acceptable level as a result of a multi-group comparative analysis. What is somewhat peculiar is that the impact of performance expectancy on technology acceptance (recommendation) intention was greater among the public than among experts. This means that the public expects that the use of carbon-neutral technologies will consequently bring about a better outcome in realizing carbon-neutral.

For perceptions about climate technologies of experts and the public, it was found that there were differences in effort expectancy, facilitating conditions, and network effects between the expert group and the public. In the public, effort expectancy and facilitating conditions were statistically significant for carbon-neutral technologies than in the expert group. It turned out that the public thought that carbon-neutral technologies were dealt with based on difficulties or conveniences, and infrastructure for carbon-neutral technologies and R&D has already been established at a certain level in Korean society. Meanwhile, network effects were significantly greater in the expert group than in the public. It is inferred that experts think that related studies and technology development achievements were somewhat lacking compared to the public or they were in progress. Consequently, to the public, it is expected that technological diffusion will advance as technology acceptance (recommendation) intention increases when policy design and promotion is made in that Korea is well-equipped with carbon-neutral technologies and that they are technologically convenient and beneficial.

Despite of valuable findings it unveiled, this study still has room for improvement. First of all, this study only focuses on the case of South Korea, which does not consider the impacts of regional or cultural differences. Hence, the reliability of the finding can be enhanced if it is supported by statistically reliable results from various countries or regions. In addition, the model can be expanded into a multi-level path analysis model considering other variables such as the dynamics of impact, the additional stages of the path, external shock, and so on. In addition, additional statistical analyses and model diversification can be applied to increase the credibility of the results. In future research, I plan to expand the scope and range of this study and design a multi-level path analysis model to solidify the theoretic model, which can be applied conventionally and enhance the statistical reliability of results.

On 18 October 2021, the Korean government virtually confirmed two items, including the 2050 Carbon-Neutral Scenario and the 2030 Plan for Raising Nationally Determined Contribution (NDC), aiming to reduce greenhouse gas emission quantity calculation by 40%

in comparison to 2018 by 2030 and achieve "net emission 0 (Net Zero)" by 2050. In addition, the government maintains related laws and systems and invests in promoting research and technology development extensively. What is important here is that government investment is consequently the tax of the people, and their support and interest are important for the laws and systems. Thus, it is most important to investigate differences in the perceptions between the expert group and the public. Therefore, this study is significant as the first study that investigated the technology acceptance and recommendation intention for carbon-neutral technologies between the expert group and the public.

**Funding:** The study was funded by the National Research Council of Science & Technology (NST) as part of a project from the NST Convergence Cluster for the Establishment of Global GreenTech Hub.

**Institutional Review Board Statement:** Not applicable.

**Informed Consent Statement:** Not applicable.

**Data Availability Statement:** The data presented in this study are available on request from the corresponding author.

**Conflicts of Interest:** The author declares no conflict of interest.

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
