# Peer review of "What Are Differences in Perceptions about Climate Technologies between Experts and the Public?"

_sustainability, doi:10.3390/su15097546_

Round 1
Reviewer 1 Report
See PDF

Reviewer 2 Report
This is an interesting article on a topic of high interest and it is my pleasure to review it. The paper is interesting and novel, but nonetheless relevant for of practitioners, policy-makers and scholars.
The paper approaches an interesting topic regarding the differences in perceptions between experts and ordinary people in terms of climate technologies, but, probably, the issue is equally important for other countries interested or engaged in the development of this technologies. Methodology and approaches are interesting, systematic and comprehensive.
The paper has merit; however, I would have some considerations and suggestions for improving the quality of the article.
Chapter 3. Methodology includes both methodological fundamentals, but also the analysis of the results and discussions. Furthermore, although there is no Chapter 4 as such, Chapter 3 is followed by the Chapter 5 Conclusion (probably a counting error). We recommend inserting this Chapter (4) of Results and discussion, restructuring the current Chapter 3 accordingly.
It is advisable, in order to make the paper clearer and more useful to the readers, as in the final part of the Results and discussion section (the section does not exist now, see the previous observation), the authors make a systematic review of the issued hypotheses, and plainly indicate the result - accepted, partially accepted, rejected etc.;
In the final part, we recommend:
- some considerations regarding the possibility of replicating the analysis, interpretations and comparability of the results in other countries, etc., it would increase the applicative value of the paper.
- a quick review of the main limitations of the paper and, hence, a proposal of further study topics, as an invitation to academic debate on proposed topics.
Formal
Figure 1. Research Model- the legend’s specifications are not found in the figure.
Thank you for the opportunity to review this article and good luck!
Reviewer 3 Report
The introduction section is too short and it does not articulate the gap (s) in literature
in a manner that will justify the study.
- The research hypothesis is not writing in the right way and it does not show the margin error and confidence level 0.95.
- The methodology section is not clear.
Reviewer 4 Report
This study addresses important questions related to the energy transition and the perception of climate technologies by experts and ordinary people. Research conducted so far in this area has rarely attempted to simultaneously analyze the perception of technology by experts and ordinary people. In a sense, this study attempts to fill an existing research gap. The great advantage of the study is the use of a balanced research sample of experts and ordinary people.
I appreciate the author efforts in the work put into this extensive research of What are Differences in Perceptions about Climate Technologies between Experts and Ordinary People? The information is easy to navigate, and the graphic structure of the paper allows readers to analyze the concepts approached, providing an interesting insight of the topic. The paper is well written according to academic standards, using proper language and scientific style.
Although, to enhance the quality of the study, it would be wise to pay attention to several issues:
1. The literature review requires a thorough reconstruction and extension with the latest items. The three most recent items (2022) presented in the article are self-citations. There is one item from 2021 and three from 2020. Therefore, it is not certain whether the research gap indicated by the author has been sufficiently identified.
2. At the end of the second section (line 233) the Research Model has been pasted. The intention of this procedure is not clear. The presented figure should be discussed in the text.
3. In point 3.3. (Analysis method) table 3 has been pasted. It would be worth commenting on the average, standard deviation, and reliability of the main variables. It is also worth noting that the list is the result of our own calculations.
4. The article also lacked an attempt to compare the results of the study with the results obtained by other researchers.
5. The interpretation of the results could be more in-depth.
6. Table 6 header formatting needs improvement.
7. In line 416 there is a reference to Table 0, which is not in the document. Probably the author wanted to refer to Table 10.
8. Interchangeable use of ‘ and “. Consistency throughout the article is recommended.
9. Extra dot on line 246.
Round 2
Reviewer 1 Report
At this revised manuscript authors proceeded in a satisfactory revision on their initial manuscript, having the reviewers’ comments addressed in a meaningful and systematic manner. In this respect the revised manuscript sustains novel characteristics and it can be accepted for publication at the Sustainability journal as is.
Reviewer 2 Report
In this new version of the article the authors have observed all our previous suggestions and recomendations.